# Deletion of Glycogen Synthase Kinase 3 Beta Reprograms NK Cell Metabolism

**DOI:** 10.3390/cancers15030705

**Published:** 2023-01-24

**Authors:** Marcelo S. F. Pereira, Kinnari Sorathia, Yasemin Sezgin, Aarohi Thakkar, Colin Maguire, Patrick L. Collins, Bethany L. Mundy-Bosse, Dean A. Lee, Meisam Naeimi Kararoudi

**Affiliations:** 1Center for Childhood Cancer and Blood Disease, Abigail Wexner Research Institute, Nationwide Children’s Hospital, Columbus, OH 43205, USA; 2Department of Microbial Infection and Immunity, The Ohio State University, Columbus, OH 43210, USA; 3Department of Internal Medicine, Division of Hematology, The Ohio State University, Columbus, OH 43210, USA; 4Department of Pediatrics, The Ohio State University, Columbus, OH 43210, USA

**Keywords:** natural killer cells, NK cells, GSK3β, AML, Cas9/RNP, CRISPR, metabolism, OXPHOS

## Abstract

**Simple Summary:**

Natural killer (NK) cells from patients with acute myeloid leukemia (AML), or from healthy donors and expanded with IL-15, show defective cytotoxicity and elevated levels of glycogen synthase kinase 3 beta (GSK3β), and drug inhibition of GSK3β was able to improve their cytotoxicity and maturation. To better understand its biologic role, we deleted *GSK3B* in NK cells and assessed their phenotype, gene expression, and function. We did not find alterations in cytotoxicity or maturation, but did observe increased metabolic function and alterations in genes related to mitochondrial metabolism. This suggests that GSK3β is a regulator of NK-cell metabolism.

**Abstract:**

Loss of cytotoxicity and defective metabolism are linked to glycogen synthase kinase 3 beta (GSK3β) overexpression in natural killer (NK) cells from patients with acute myeloid leukemia or from healthy donors after expansion ex vivo with IL-15. Drug inhibition of GSK3β in these NK cells improves their maturation and cytotoxic activity, but the mechanisms of GSK3β-mediated dysfunction have not been well studied. Here, we show that expansion of NK cells with feeder cells expressing membrane-bound IL-21 maintained normal GSK3β levels, allowing us to study GSK3β function using CRISPR gene editing. We deleted *GSK3B* and expanded paired-donor knockout and wild-type (WT) NK cells and then assessed transcriptional and functional alterations induced by loss of GSK3β. Surprisingly, our data showed that deletion of *GSK3B* did not alter cytotoxicity, cytokine production, or maturation (as determined by CD57 expression). However, *GSK3B*-KO cells demonstrated significant changes in expression of genes related to rRNA processing, cell proliferation, and metabolic function, suggesting possible metabolic reprogramming. Next, we found that key genes downregulated in *GSK3B*-KO NK cells were upregulated in GSK3β-overexpressing NK cells from AML patients, confirming this correlation in a clinical setting. Lastly, we measured cellular energetics and observed that *GSK3B*-KO NK cells exhibited 150% higher spare respiratory capacity, a marker of metabolic fitness. These findings suggest a role for GSK3β in regulating NK cell metabolism.

## 1. Introduction

The antitumor activity of NK cells has been widely demonstrated for multiple cancer types [1]. This antitumor role can be limited by several factors in the tumor microenvironment that mediate metabolic suppression [2,3]. As a result, chemical inhibitors, genetic, and epigenetic alterations are now being tested to overcome these suppressive mechanisms in NK cells [4]. One promising target to enhance NK cell cytotoxic activity is GSK3β, a serine threonine kinase. GSK3β has been shown to regulate multiple functions in other cell types, but its role in lymphocytes including NK cells has not been well studied. In a few reports, drug inhibition of GSK3β has been shown to improve maturation and antitumor activity of NK cells with elevated GSK3β, such as IL-15 expanded NK cells or NK cells isolated from AML patients [5,6,7].

We recently described an approach for studying human NK cell biology by combining NK cell expansion with CRISPR gene editing using Cas9 complexed with guide ribonucleoproteins (Cas9/RNP). Here, we apply this approach to studying the transcriptional and functional role of GSK3β by generating CRISPR mediated *GSK3B*-KO NK cells, to avoid the confounding off-target effects of small-molecule drug inhibition. To confirm the clinical relevance of the differentially expressed genes found in *GSK3B*-KO NK cells, we looked at expression of these genes in NK cells from patients with AML, which we previously showed to have elevated GSK3β [8,9]. Lastly, we investigated the metabolic alterations in NK cells induced by loss of GSK3β.

## 2. Material and Methods

### 2.1. Source of Primary Human NK Cells and Regulatory Approvals

Buffy coats from healthy volunteer red blood cell donations were obtained from the American Red Cross (Columbus, OH, USA). Peripheral blood was obtained from patients with untreated AML (Ohio State University Leukemia Tissue Bank). All studies and sample collections were approved through the Ohio State University Institutional Review Board, protocol number 2009C0019, or deemed IRB exempt.

### 2.2. Purification and Expansion of NK Cells

NK cells from AML patients and controls from normal donors were isolated by density-gradient purification followed by flow-based sorting. NK cells from both healthy donors and untreated AML patients were sorted into two subpopulations based on NK cell development (CD94^+/-^/NKp80^+^/CD16^+^/CD57^-^ and CD94^+/-^/NKp80^+^/CD16^+^/CD57^+^ (stage 5 and 6, respectively)) as previously described [10,11,12,13,14,15]. NK cells for in vitro gene editing studies were isolated by negative selection using RosetteSep™ Human NK Cell Enrichment Cocktail (Stem Cell Technologies, 15065, Vancouver, BC, Canada). Purified NK cells (depleted of T cells, B cells, and monocytes and typically > 95% CD16/56+) were stimulated weekly for two weeks with irradiated CSTX002 feeder cells in AIM-V expansion medium supplemented with ICSR (CTS™AIMV™SFM/CTS^TM^ Immune Cell SR, Thermo Fisher Scientific) [16] and 50 IU of human recombinant IL-2 (rIL-2) (Novartis). NK cell purity of WT and KO NK cells was determined after expansion and was uniformly > 98% (Appendix A).

### 2.3. Tumor Cell Lines

HL60 (AML), Kasumi1 (AML), K562 (CML), DAOY (medulloblastoma), and MG63 (osteosarcoma) cell lines were purchased from the American Type Culture Collection (ATCC, Manassas, VA, USA). U373 (glioblastoma) was kindly shared by Kevin Cassady (Nationwide Children’s Hospital, Columbus, OH, USA). CSTX002 feeder cells (K562 genetically modified to express 4-1BBL and membrane-bound IL-21, referred to hereafter as FC-21) was generated as previously described [16].

### 2.4. Generation of CRISPR-Edited NK Cells

*GSK3B*-KO NK cells were generated by electroporation of Cas9/RNP into NK cells at day 7 of expansion [17,18,19], targeting exon 5 of the *GSK3B* gene (5- CAGTATCAGGATCCAACAAG) as described previously [17,18]. Wildtype (WT) cells were electroporated in electroporation buffer without Cas9/RNP complexes as control.

### 2.5. NK Cell Functional Assays

Calcein-AM was used to evaluate cytotoxicity previously described [20]. Briefly, tumor cell targets were loaded with 2 ug/mL of Calcein-AM for 30 min. Cells were washed and incubated with WT or *GSK3B*-KO NK cells at multiple effector/target (E:T) ratios for 4 h. For long-term killing assays, real time cell analysis—(RTCA) was performed. Briefly, the xCELLigence SP instrument (Agilent) was utilized to record cellular impedance following the previously described [21,22]. To analyze killing of target cells by NK cells, the E-Plate wells were incubated with the tethering reagent CD29 for Kasumi cell line. WT or *GSK3B*-KO NK cells—effector cells—were added on top of the tumor cells at 2:1 (Effector: Target) ratio. Each condition was tested in triplicate and total of three donors were evaluated. An “effector cell only” control was included to demonstrate that NK cells do not attach in presence of anti-CD29 and subtracted from the equation. Cytotoxic activities were evaluated based on the viability of the tumor cells remained attached in E-Plate surface, as reflected by Cell Index values. The xIMT software was used to plot the percentage of cytolysis. Experiment was conducted in RPMI1640 medium with 10% FBS supplemented with 50 IU of IL-2.

### 2.6. Metabolic Assays

Metabolic assays were performed as previously described [17]. Briefly, we used Seahorse XF Cell Mito Stress Test Kit to measure the oxygen consumption rate (OCR) (Cat# 103015-100, Agilent Technologies, Santa Clara, CA, USA) and Seahorse XF Glycolysis Stress Test Kit to measure extracellular acidification rate (ECAR) (Cat# 103020-100, Agilent Technologies, Santa Clara, CA, USA).

### 2.7. Antibodies

The following antibodies were used for flow cytometry: NCAM-1 (CD56) (Cat#744217—BD), CD57 (Cat#130-111-964—Miltenyi Biotec). The following antibodies were used for Western Blot: GSK3β Rabbit mAb—27C10 (Cat#9315S—Cell Signaling—1:1000), ß-Actin Mouse mAb—8H10D10 (Cat#3700S—Cell Signaling—1:1000), Anti-Rabbit IgG HRP-linked (Cat#7074S—Cell Signaling—1:5000), Anti-Mouse IgG HRP-linked (Cat#7076S—Cell Signaling—1:5000).

### 2.8. Cytokine Secretion

To induce cytokine secretion, NK cells at 2 × 10^6^/mL were stimulated with 10 μg/mL PHA. After 4 h of incubation, supernatants were collected and kept in −80 °C. On the day of the assay, the supernatants were thawed and measured in duplicate with Bio-Rad Bio-plex Pro Human Immunotherapy Panel 20-Plex (Cat#12007975) according to manufacturer’s instructions. Data were acquired on Bio-rad Bio-Plex 200 system and analyzed with Bio-plex Manager software using curve-fitting with logistic regression (5PL regression) [23].

### 2.9. RNA-Sequencing on Non-Expanded Healthy and AML-NK

RNA-sequencing (RNA-seq) analysis was done as previously described [10]. Briefly, freshly sorted NK cell subpopulations from normal donor peripheral blood (American Red Cross; *n* = 3 donors) or newly diagnosed AML patients (Ohio State University Leukemia Tissue Bank; n = 5; OSU IRB# 2009C0019), were pelleted and total RNA was isolated using the Qiagen RNeasy Mini Kit (Qiagen). Directional poly-A RNA sequencing libraries were prepared and sequenced as 42-bp paired-end reads on an Illumina NextSeq 500 instrument (Illumina) to a depth of 33.2–48.0 × 10^6^ read pairs (Active Motif). Alignment to human genome (hg19 build) was done using TopHat [24]. Transcriptome assembly and analysis was performed using Cufflinks and expression was reported as FPKM.

### 2.10. RNA-Sequencing on Expanded WT and GSK3B-KO NK Cells

For RNA-seq on expanded WT and *GSK3B*-KO NK cells, RNA libraries were prepared using the TruSeq RNA Sample Preparation Kit (Illumina Inc. San Diego, CA, USA) and sequence reads (150 bp each) were generated per library using the Illumina HiSeq4000 platform (Institute for Genomic Medicine, Nationwide Children’s Hospital). Reads were aligned and count tables were generated using Kallisto (v 0.43.1) [25]. Differential expression was then done using the Bioconductor package DeSeq2 (v 1.36.0). Volcano plots were generated using the package Glimma (v. 2.6.0) [26] within R. Gene ontology (GO) was performed using Gorillia [27] and subsequently visualized using REVIGO [28] via their online web portal.

## 3. Results

### 3.1. Deletion of GSK3B in FC-21 Expanded NK Cells

We previously described a gene editing approach for primary NK cells [17,18] in which propagation with FC-21 feeder cells enhances DNA repair machinery and expands edited cells to large numbers [19]. It was previously shown that *GSK3β* becomes overexpressed when NK cells are expanded in presence of soluble IL-15 and is associated with inhibition of their maturation and cytotoxic activity [5]. Therefore, we first evaluated the relative gene expression of *GSK3B* in both naïve and FC-21-expanded NK cells to assess the stability of *GSK3β* in this model system. We found that the expression level of *GSK3B* was similar between WT and expanded NK cells (Figure 1A).

To study the role of GSK3β and to avoid off-target effects of chemical inhibitors we generated *GSK3B*-KO NK cells and evaluated the knock-out efficiency at the protein level by Western blot (Figure 1B, uncropped blots shown in Appendix A), and at the mRNA level by RNA-sequencing (Figure 1C) observed as a read drop across the region targeted by the gRNA at exon 5 of the *GSK3B* locus. Our data demonstrated successful *GSK3β* gene deletion in FC-21 expanded NK cells.

### 3.2. Deletion of GSK3B Does Not Alter Proliferation, Killing Potency, Cytokine Secretion, or Maturation of FC-21 Expanded NK Cells

Drug inhibition of GSK3β can improve NK cell killing against AML in NK cells with elevated *GSK3β,* such as those expanded with IL15 [5] or from patients with AML [7]. Therefore, we next assessed the effect of knocking out *GSK3B* in FC-21 expanded NK cells on their anti-AML activity. Unexpectedly, the absence of GSK3β did not alter NK cell proliferation (Appendix A) or killing against Kasumi1, an AML cell line, in either short (Figure 2A) or long-term (Figure 2B,C) killing assays. Additionally, *GSK3B*-KO NK cells showed similar killing potency as WT NK cell against HL60 (AML), K562 (CML), DAOY (medulloblastoma), and MG63 (osteosarcoma) in a standard 4 h killing assay (see Appendix A). In the previously reports, the improved killing in NK cells treated with GSK3β chemical inhibitors was also associated with increased cytokine secretion. Thus, we evaluated cytokine secretion in WT and *GSK3B*-KO and found that both WT and *GSK3B*-KO NK cells secrete high levels of IL-2, IFN*γ* and TNF*α*, and no difference was found between the two (Figure 2C).

Additionally, Cichocki et al., previously reported that GSK3β drug inhibition drove maturation of IL-15 expanded NK cell, evidenced by CD57 expression [5]. Thus, we analyzed CD57 expression on both FC-21 expanded WT and *GSK3B*-KO NK cells by flow cytometry and found no difference between them (Figure 2D). We then investigated FC-21 expanded NK cells for their expression of *B3GAT1*, the gene encoding the enzyme responsible for catalyzing the CD57 carbohydrate epitope. Interestingly, we found that *B3GAT1* was 30-fold downregulated in FC-21 expanded NK cells compared to naïve NK cells, and that this low expression was not altered by *GSK3B* deletion (Figure 2E). This explains the low CD57 surface expression in both WT and gene edited cells, and also suggests that *B3GAT1* is not directly regulated by GSK3β and that other elements may be involved in regulating the expression of CD57 in NK cells. 

### 3.3. Transcriptional Changes in GSK3B-KO NK Cells

To study the role of GSK3β in NK cells, we next performed bulk RNA-seq on WT or *GSK3B*-KO NK cells after expansion with FC-21. Differential gene expression analysis via Deseq2 revealed 55 genes (12 upregulated and 43 downregulated) with significantly altered expression (adjusted *P* value < 0.05, Paired Deseq2 test) in WT NK cells compared to *GSK3B*-KO cells. The most significant of these differentially expressed genes was GSK3B (*P* = 1.59 × 10^−14^), thus the majority of changes detected were of loss of expression concurrent with the loss of GSK3β expression (Figure 3A,D).

We next used gene ontology (GO) analysis of the 55 WT-specific transcripts to determine what expression programs were regulated by GSK3β. GO results were then visualized via REVIGO, which clusters redundant categories for biological interpretation (Figure 3B). Unique GO clusters included those for ribosomal RNA (rRNA) processing, secretion, metabolic processes, and proliferation (Figure 3B). For example, *GSK3B*-KO lost transcription of genes involved within NK cell homeostatic renewal, such as the high affinity IL2 trimeric receptor, encoded by *IL2RA*, and the survival gene *BCL2* (Figure 3C). We likewise observed downregulation of genes that encode for TNFα effectors, namely leukemia inhibitory factor (*LIF*) and *TNFSF10* (encoding TRAIL), within *GSK3B*-KO. Additional expression analysis suggested deregulation of global biosynthesis as some genes involved in ribosomal RNA processing were lost (e.g., *WDR3*, *WDR74*, see Figure 3C) and those involved in mitochondrial function were gained (e.g., *MT-ND4* and *MT-ND2*, see Figure 3C). Together, transcriptome analysis suggested a role of GSK3β in regulating NK cell homeostasis, effector function, and metabolism.

### 3.4. Deletion of GSK3B Leads to Metabolic Reprogramming of NK Cells

To explore the effect of upregulation of the mitochondrial genes identified by RNA-seq, we next examined the cellular metabolism of WT and *GSK3B*-KO NK cells by assessing both mitochondrial (Figure 4A,D,E) and glycolytic stress (Figure 4B,F–H). We found that *GSK3B*-KO NK cells presented an overall shift towards oxidative metabolism with higher OCR and spare respiratory capacity (Figure 4A–E). Spare respiratory capacity was previously described as pivotal for memory T cell formation and as a reliable indicator of metabolic fitness [29,30]. Additionally, *GSK3B*-KO NK cells also showed modestly higher ECAR, glycolytic capacity, and glycolytic reserve than WT NK cells, indicating that the increase in oxidative phosphorylation was not compensatory for decreased glycolysis. Together this suggests that deletion of *GSK3B* induces NK cells to enhance overall metabolic capacity, which could be beneficial for sustaining antitumor activity.

### 3.5. GSK3β Is Highly Expressed in NK Cells at Stage 5 and 6 of Maturation from AML Patients

To further study the role of GSK3β in NK cells, we studied transcriptional changes in NK cells from AML patients which have been shown to overexpress GSK3β, to validate key genes identified as differentially expressed in *GSK3B-KO* NK cells. To correct for confounding by differences in NK cell maturation, we separately studied the expression of level of these genes in NK maturation stages 5 and 6. First, we confirmed that *GSK3β* was statistically elevated in NK cells from AML patients when compared to healthy donor NK cells, for both stage 5 and stage 6 cells. We then evaluated the relative expression of the key genes identified in *GSK3B-KO* NK cells (Figure 5) and found strong concordance of downregulated genes in *GSK3B-KO* NK cells being upregulated in *GSK3β*-overexpressing NK cells from patients with AML. *LIF* was only overexpressed in stage 5 but not in stage 6. *IL2RA* expression was significantly increased in both AML-NK5 and AML-NK6 when compared to healthy NK cells. No difference was observed in expression level of *WDR74,* but *WDR3* was significantly increased in AML-NK5. We did not identify any mRNA transcripts for *MT-ND4* and *MT-ND2* in any of these samples, despite finding MT-ND4 and MT-ND2 transcripts at high levels in the *GSK3B-KO* experiments. Since these mtDNA transcripts are not recovered efficiently by all purification methods, this may be explained differences in RNA isolation and sequencing methods in the *GSK3B-KO* and AML-NK experiments. Although the reverse correlation between the differentially expressed genes in *GSK3B*-KO NK cells versus AML-NK cells suggests a possible role of *GSK3β* on NK-cell activity in these patients, further studies would be beneficial.

## 4. Discussion

NK cells play an important role in immunosurveillance and preventing tumor development and progression. NK cells from patients with AML exhibit great defects in both numbers and functional activities, unable to control AML development, progression and relapse [31,32]. Even though phenotypic changes have been well described, specific molecular explanations for these dysfunctions are still needed [33]. It was previously reported that the high expression of GSK3β on AML NK cells profoundly impacted NK cells killing ability [7]. Here, we further demonstrated that the enhanced expression of GSK3β is present in NK cells at both stage 5 and 6 of development in patients with AML. There was previously limited evidence on the biologic mechanisms of GSK3β-mediated dysfunction in NK cells. Here, we used Cas9/RNP gene editing to generate *GSK3B*-KO NK cells [17,18,19] to enable precise gene-level understanding of GSK3β effects on primary human NK cells, which revealed novel information on the impact of GSK3β on transcriptional regulation of homeostasis, effector function, and metabolism.

We showed that expansion of NK cells on FC-21 does not increase GSK3β expression levels as was observed in IL15-expanded NK cells, which may reduce the need for GSK3β inhibition in adoptive NK cell therapy.

Deletion of *GSK3B* resulted in increased mitochondrial respiratory capacity that was associated with an increase in Complex 1 genes, MT-ND2 and MT-ND4, which favor OXPHOS metabolism. Defined as “The Most Complex Complex” [34], Complex 1, also known as NADH dehydrogenase, is composed of 46 subunits, seven of which (including MT-ND2 and MT-ND4) are encoded in the mitochondrial genome. Complex 1 catalyzes electrons transfer from NADH through the respiratory chain, using ubiquinone as an electron acceptor [35]. Furthermore, the expression of mitochondrial respiratory complex 1 was previously associated with gain in mitochondrial oxidative phosphorylation (OXPHOS) activity. Additionally, targeting OXPHOS with a complex I inhibitor decreased OXPHOS in pancreatic cancer cells reinforcing the correlation between OXPHOS and Complex 1 [36]. In line with previously published evidence, our data showed that *GSK3B* deletion resulted in increased OXPHOS, with higher maximum and spare respiratory capacities. The edited cells also shift their metabolic profile more towards OCR than ECAR which can improve ATP generation and provide more energy to the cells essential for effector function. Previously Bou-Tayeh et al. [37], demonstrated that NK cells from both leukemic mice and patients with AML displayed similar metabolic defects.

Several groups showed that NK cells from AML patients or expanded with IL15 have high levels of GSK3β and that chemical inhibition of GSK3β can improve their antitumor activity through an increase in NFKB signaling molecules (*RELA, RELB, c-REL, NFKIBA*) [5] However, our data showed that FC21 expansion of NK cells does not increase GSK3β levels, and therefore its deletion did not alter their cytotoxicity, in both short and long-term killing assays we used here. In vivo experiments or other approaches that mimic the harsh tumor microenvironment may be able to show the effect of these metabolic improvements on anti-tumor activity of *GSK3B-KO* NK cells.

NK cells with high GSK3β have been shown to express low levels of CD57. A benefit of using GSK inhibitors in these NK cells was to increase CD57 expression level suggesting maturation to an adaptive memory-like phenotype. This protein is recognized as a maturation/senescence marker and its frequency increases with age [36] and it has been reported to be missing or dim on fetal and infant NK cells [38]. The CD57 expression, denoting terminal differentiation of NK cells, should be beneficial from an immunotherapy standpoint, as several clinical studies reported that high expression of CD57 on NK cells is associated with better clinical outcome [5,38,39]. It has been shown that drug inhibition of GSK3β resulted in increased CD57 expression. Of note, CD57 is not a DNA-encoded protein, but rather a carbohydrate epitope catalyzed by B3GAT1 [40]. We showed that B3GAT1 levels decrease in FC-21 expanded NK cells, but that both CD57 and B3GAT1 remain unchanged after the deletion of *GSK3B*. This suggests that GSK3β does not directly regulate B3GAT1, or therefor CD57, implying other regulators and/or pathways control the CD57 expression associated with maturation and memory. We also note that there was no correlation between *GSK3B* expression and Stage 5 or 6 NK cells, for AML-NK or healthy donor NK cells, despite Stage 5 and 6 being defined by the expression of CD57.

## 5. Conclusions

Using CRISPR gene deletion of *GSK3B* in human primary NK cells we showed that the *GSK3B* does not any affect in cytotoxicity or maturation of the cells, but has a role in metabolic function in NK cells. Taken together, we show that GSK3β is a negative metabolic regulator in NK cells.

## Figures and Tables

**Figure 1 cancers-15-00705-f001:**
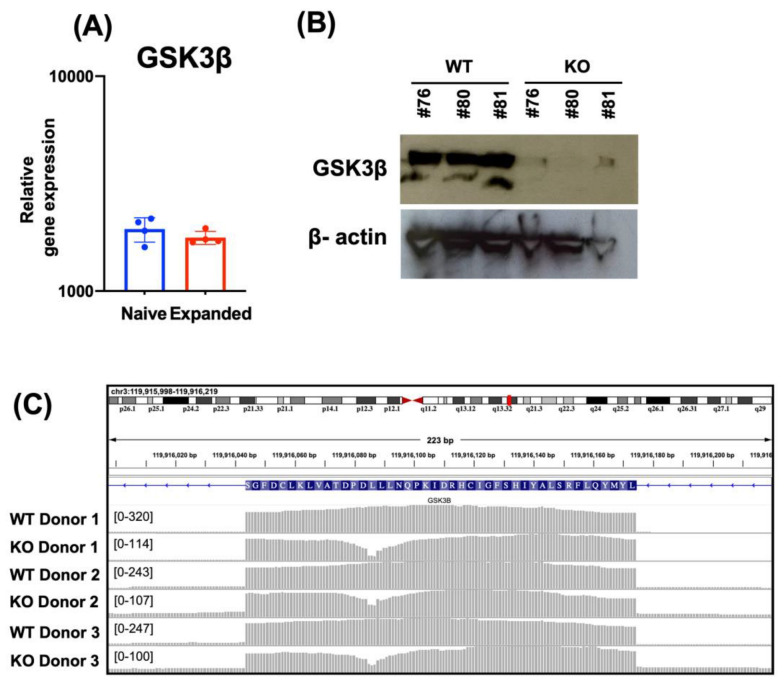
Generation of *GSK3B*-KO FC-21 expanded NK cells. *GSK3B* expression in NK cells before and after expansion on FC-21 was evaluated by RNAseq analysis. (**A**) The efficiency of Cas9/RNP mediated deletion of *GSK3β* was determined by Western blot (**B**) and RNA-seq (**C**).

**Figure 2 cancers-15-00705-f002:**
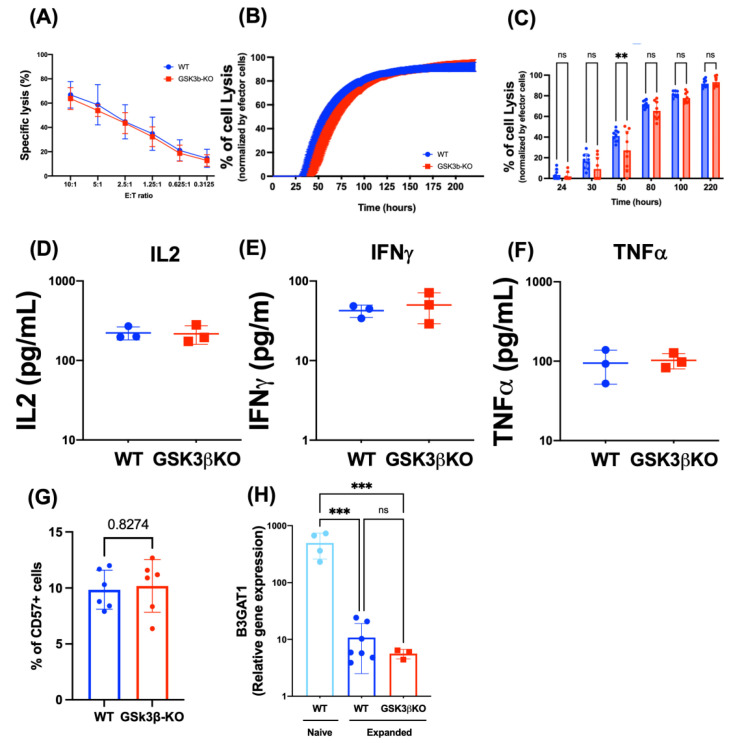
Deletion of *GSK3B* did not alter antitumor activity or maturation of FC-21 expanded NK cells. Both WT and *GSK3B*-KO NK cells were cocultured with Kasumi1 (AML tumor cell line) for 4 (**A**) or 220 h (**B**,**C**). “B” time course of long-term cell killing and “C” bar graph representative of indicated hours; (*n* = 3). (**D**–**F**) IL2, IFN*γ* and TNF*α* release was assessed after 4 h of WT and *GSK3B*-KO NK cells with PHA (*n* = 3). (**G**), CD57 expression was evaluated by flow cytometry on FC-21 expanded WT and *GSK3B*-KO NK cells (*n* = 6). (**H**), B3GAT1 expression in WT, FC-21 expanded WT, and FC-21 expanded *GSK3B*-KO NK cells as measured by RNA-seq. ns = non-significative, ** *p* ≤ 0.01, *** *p* ≤ 0.001. See also Appendix A for cytotoxicity against other cancer cell lines, and Appendix A for representative CD57 gating.

**Figure 3 cancers-15-00705-f003:**
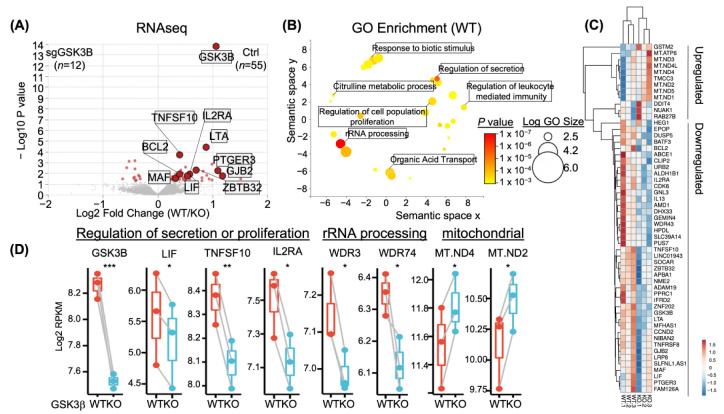
*GSK3B*-KO in NK resulted in transcriptomic changes, RNA-seq analysis was performed on NK cells from FC-21 expanded WT and *GSK3B*-KO. (**A**) volcano plot representing DEG (WT/KO). (**B**) GO enrichment (WT). Heatmap (**C**) and RNA-seq boxplots (**D**) showing the top downregulated and top upregulated genes affected by *GSK3B* deletion. * *p* ≤ 0.05, ** *p* ≤ 0.01, *** *p* ≤ 0.001. For (**C**), rows are centered; unit variance scaling is applied to rows. Both rows and columns are clustered using correlation distance and average linkage.

**Figure 4 cancers-15-00705-f004:**
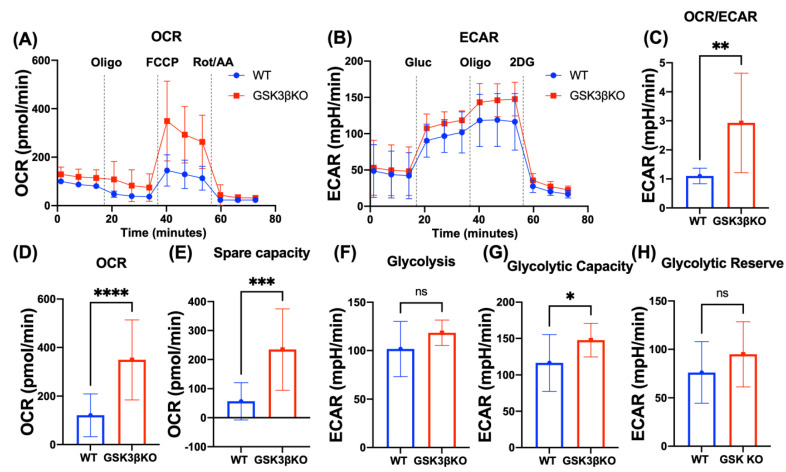
Deletion of *GSK3B* enhances metabolic capacity of NK cells. (**A**,**B**—mitochondrial and glycolytic stress, respectively) summarized data of metabolic analysis of paired WT and *GSK3B*-KO NK cells (*n* = 3; mean ± SD). (**C**) OCR/ECAR ratio. (**D**) Maximum respiratory capacity. (**E**) reserve capacity derived from (**A**). (**F**) Glycolysis, (**G**) glycolytic capacity, and (**H**) glycolytic reserve derived from (**B**). *n* = 4, ns = non-significant, * *p* ≤ 0.05, ** *p* ≤ 0.01, *** *p* ≤ 0.001, **** *p* ≤ 0.0001.

**Figure 5 cancers-15-00705-f005:**
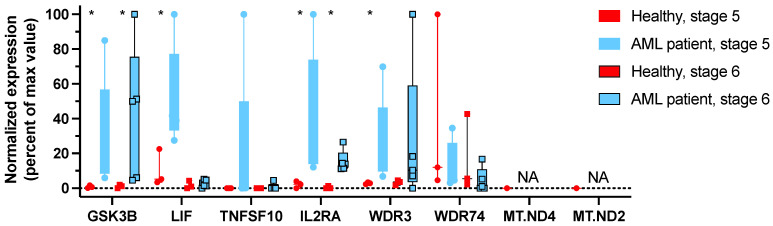
The downregulated genes identified in *GSK3B*-KO NK cells were mostly upregulated in GSK3β-overexpressing NK cells from patients with AML. RNA-seq analysis was performed on NK cells at different stages of maturation isolated by sorting from healthy donors and patients with AML. * *p* ≤ 0.05. (Healthy donors, *n* = 3; AML patient, *n* = 5).

## Data Availability

All data reported in this paper will be shared by the correspondence authors upon request.

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
