# Peer review of "Deletion of Glycogen Synthase Kinase 3 Beta Reprograms NK Cell Metabolism"

_cancers, 2023, doi:10.3390/cancers15030705_

Round 1

Reviewer 1 Report (Previous Reviewer 1)

The authors showed their readiness by conducting 2 additional experiments. Unfortunately they used a different IL-15 culture model as the one used in Cichocki, and they did not succeed to delete the GSK3B gene in NK cells from AML patients.

Reviewer 2 Report (Previous Reviewer 3)

Dear Authors

I highly appreciate the revised manuscript. The issues pointed in the previous version have been resolved. I have no further comments.

This manuscript is a resubmission of an earlier submission. The following is a list of the peer review reports and author responses from that submission.

Round 1

Reviewer 1 Report

It has been previously shown that a small molecule inhibitor of GSK3β improves maturation and cytotoxic antitumor activity of NK cells with elevated GSK3β, such as IL-15 expanded NK cells or NK cells isolated from AML patients. In this report, the authors combine NK cell expansion with CRISPR gene editing using Cas9/RNP to generate GSK3β-KO NK cells. The aim was “to study the transcriptional and functional role of GSK3β without possible confounding off-target effects of small- molecule drug inhibition”. To expand the NK cells prior to CRISPR/Cas9 treatment, NK cells were cultured on FC-21 feeder cells. It is shown that GSK3β expression is similar between naive and expanded NK cells, and that GSK3β deletion does not alter antitumor activity or maturation of FC-21 expanded NK cells. The transcriptional changes in GSK3β-KO versus WT NK cells was determined by RNA-seq analysis. It is experimentally shown that deletion of GSK3β induces NK cells to enhance overall metabolic capacity, and it is suggested that this could be beneficial for sustaining antitumor activity. Finally, it is shown that GSK3β is highly expressed in stages 5 and 6 NK cells from AML patients, and that 3 out of 7 genes that were shown to be differentially expressed in GSK3β KO NK cells were differentially expressed in the opposite direction in stage 5 AML NK cells.

Major

One of the aims of this study was to study whether the increased GSK3β expression in IL-15 expanded NK cells and in AML NK cells is the cause of their decreased maturation and cytotoxic antitumor activity, as previously indicated by the effects of a small molecule inhibitor of GSK3β. It is therefore unfortunate that a culture model was used here that did not show an increase in GSK3β expression in NK cells.

It should be indicated whether the WT NK cells, as the GSK3β KO NK cells, underwent the whole electroporation process, thus electroporation in the presence of CRISPR and Cas9, but in the absence of guide ribonucleoproteins. If WT NK cells have not undergone such a control electroporation process, this may result in differential gene expression compared to GSK3β KO NK cells that is not due to GSK3β.

The manuscript would be strengthened by showing that the increased metabolic capacity of GSK3β KO NK cells has effects on proliferation, survival and/or (sustained) cytotoxicity. The first 2 aspects are not addressed. Regarding cytotoxicity, it is shown that there is no difference in killing capacity of GSK3β KO compared to WT NK cells in a 4 h killing assay. One might consider coculturing NK cells with tumor target cells at a low E:T ratio during 24 or 48 h and then determining the number of remaining viable tumor cells.

The similarity (opposite direction) between the differential gene expression in GSK3β KO NK cells versus NK cells from AML patients is rather limited and thus the conclusion should be tempered.

Minor

It should be indicated in Mat&Meth and in the Results section how many replicates were used for the RNA-seq analysis. Also, the number of sequence reads (line 119) should be indicated.

Line 153-154 (“Additionally, GSK3β-KO NK cells showed similar killing potency …”): are these “data not shown”? Please indicate.

Figure 3:

There is no “A”, “B” or “C” labeling in the figure

It should be indicated in the X-axis label of Fig. 3A which Log Fold Change is shown. Log2 Fold Change?

The Y-axis of Fig. 3C reads “Min – Max”. It needs to be explained how this should be interpreted. What does “Min” and “Max” mean? The Y-axis also needs numbers to allow interpretation of the degree of difference in WT vs KO.

There is a typo in the heading “TNFSF10”

Figure 4:

The X-axis of A) and B) goes from 0 to 80. Are these seconds, minutes?

It should also be indicated at which time point oligomycin, FCCP, rotenone, and antimycin A were added.

Figure 5:

It has to be explained what is meant with “percent of max value” in the Y-axis. How is it possible that most values of healthy NK cells are almost zero?

There are several typos in the manuscript. These should be corrected.

Author Response

Reviewer 1:

  • One of the aims of this study was to study whether the increased GSK3β expression in IL-15expanded NK cells and in AML NK cells is the cause of their decreased maturation and cytotoxicantitumor activity, as previously indicated by the effects of a small molecule inhibitor of GSK3β. It is therefore unfortunate that a culture model was used here that did not show an increase in GSK3β expression in NK cells

We thank you for your comment. In the line 22-24, we clarified that FC-21 expanded NK cells were used as a  model to study the biology of GSK due their stable expression of GSK after expansion. Using this model we could genetically engineer NK cells and we didn’t see any increase in the GSK3B level. Using models with elevated GSK3b such as IL15 expanded NK cells with would not be optimal to study the biology of these cells and it has also been studied by Miller lab at Cichocki et al, 2017.

  • It should be indicated whether the WT NK cells, as the GSK3β KO NK cells, underwent the wholeelectroporation process, thus electroporation in the presence of CRISPR and Cas9, but in the absence of guide ribonucleoproteins. If WT NK cells have not undergone such a control electroporation process, this may result in differential gene expression compared to GSK3β KO NK cells that is not due to GSK3β.

We thank you for your comment. We added a description in the lines 80-81, clarifying the methodology. We indeed electroporated WT NK cells in the absence of Cas9/RNP complex.

  • The manuscript would be strengthened by showing that the increased metabolic capacity of GSK3βKO NK cells has effects on proliferation, survival and/or (sustained) cytotoxicity. The first 2 aspects are not addressed. Regarding cytotoxicity, it is shown that there is no difference in killing capacity of GSK3β KO compared to WT NK cells in a 4 h killing assay. One might consider coculturing NK cells with tumor target cells at a low E:T ratio during 24 or 48 h and then determining the number of remaining viable tumor cells.

We thank you for your comment. We performed requested experiments and the results were added to the manuscript  and the material/methods and the results section (see figure 2 B and C). However, no significant changes were observed.

  • The similarity (opposite direction) between the differential gene expression in GSK3β KO NK cellsversus NK cells from AML patients is rather limited and thus the conclusion shouldbe 

We thank you for your comment. We added one sentence in the lines 364-382 to reflect your valuable comment.

Minor

  • It should be indicated in Mat&Meth and in the Results section how many replicates were used forthe RNA-seq  Also, the number of sequence reads (line 119) should be indicated.

We thank you for your comment. The number of reads are now indicated in lines 173-174.

  • Line 153-154 (“Additionally, GSK3β-KO NK cells showed similar killing potency …”): are these “datanot shown”? Please indicate.

We thank you for your comment. The original submitted manuscript contained the requested data, but it seems that it was not sent to the reviewers. You may find it now in the new version.

  • Figure3 and 4 :

We thank you for your comment. All the requested changes were applied.

  • Figure5: It has to be explained what is meant with “percent of max value” in the Y-axis. How is it possible that most values of healthy NK cells are almost zero?

We thank you for your comment. The values for expression of each gene were normalized to the maximum expression level detected among all of the samples. This shows that these genes were induced or overexpressed in NK cells from patients with AML, and appear low or non-expressed in NK cells from healthy donors by comparison.

Reviewer 2 Report

Significant revision is required to address the following critiques:

1.       Fig 1. Please provide better labels and legends. As in its current, it is very difficult to understand the data presented in Fig 1a and 1c.

2.       Fig 2. What is the killing ability of naïve NK in comparison to the functional readout of 2A-2C?

3.       Fig 2B – how to explain the WT-NK killing is>100%?

4.       Fig 3- could the authors please use a heatmap to present the top 20 upregulated and downregulated genes with GSK3b knockout?

5.       Fig 5. The authors need to provide validation data in Fig 5.

6.       Fig 5 – needs to indicate patient number n in the Figure legend.

7.       Authors please elaborate the significance of the findings in the abstract

Author Response

Reviewer 2:

  • Please provide better labels and legends. As in its current, it is very difficult to understand the data presented in Fig 1a and 1c

We thank you for your comment. All the requested changes were applied to the figures.

  • What is the killing ability of naïve NK in comparison to the functional readout of 2A-2C?

We thank you for your comment. Based on our several publications such as :0.1371/journal.pone.0030264, using naïve (freshly isolated, non-expanded) NK cell would not be a good control to include due to their significantly lower killing ability. Therefore, we did not use the naïve NK cells but instead used expanded WT NK cells. We cited our previous works throughout the manuscript

  • how to explain the WT-NK killing is>100%?

We thank you for your comment. The slight increase ~+10% is due to technical issue commonly seen during the Calcein-AM release assay with some cancer lines. This can be seen in both groups (WT and KO).

  • could the authors please use a heatmap to present the top 20 upregulated and downregulated genes with GSK3b knockout?

We thank you for your comment. All the requested changes were applied.

  • The authors need to provide validation data in Fig 5

We thank you for your comment. We were not sure which validation methodology we had to use to address your question. However, we were not able to run any other experiments using thes rare primary NK cells.

  • needs to indicate patient number n in the Figure legend.

We thank you for your comment. All the requested changes were applied

Reviewer 3 Report

The main objective of this paper is to decipher the role of GSK3B in NK cell functions in patients with acute myeloblastic leukemia. Indeed, GSK3B is elevated in NK cells from AML patients and drug inhibition of GSK3B improves their cytotoxic activity.

This work includes the development of a GSK3B KO with Cas9/RNP targetting exon5, valided at both RNA and protein level.

With an in-vitro model, the transcriptomic analysis shows a limited set of differencially expressed genes beetwen WT and GSK3B-KO, mainly related to NK cell effector function. Among 12 downregulated genes in GSK3B-KO, the authors identified 2 genes involved in mitochondrial function and suggested a role of GSK3B in metabolism. Unfortunately, they did not show any difference on NK cell phenotype, cytotoxicity or cytokine production, altering the global message of this paper.

They did not compared the transcriptomic profile of NK cells from healthy donnor vs AML patients, but only pointed the expression of the genes selected from their in-vitro experiments.

Lastly, the conclusions on the role of GSK3B in AML patients are overinterpreted.

This work is nicely conducted and includes many interesting experiments, but the main message is not well supported by the functional analyses.

Minor revision:  The order and legends of figures are sometimes approximative.  

Please find inclosed the paper with my points of discussion.

Sincerely,

Author Response

Reviewer 1:  2nd round comments

My first major comment was: “One of the aims of this study was to study whether the increased GSK3β expression in IL-15-expanded NK cells and in AML NK cells is the cause of their decreased maturation and cytotoxic antitumor activity, as previously indicated by the effects of a small molecule inhibitor of GSK3β. It is therefore unfortunate that a culture model was used here that did not show an increase in GSK3β expression in NK cells.”

The answer of the authors is: “We thank you for your comment. In the line 22-24, we clarified that FC-21 expanded NK cells were used as a model to study the biology of GSK due their stable expression of GSK after expansion. Using this model we could genetically engineer NK cells and we didn’t see any increase in the GSK3B level. Using models with elevated GSK3b such as IL15 expanded NK cells with would not be optimal to study the biology of these cells and it has also been studied by Miller lab at Cichocki et al, 2017.”

This is an evasive and irrelevant response. As indicated by the authors themselves in this manuscript, one of the aims was to study whether the increased GSK3β expression in IL-15-expanded NK cells and in NK cells from AML patients is causing their decreased maturation and cytotoxic antitumor activity. Evidence for this was provided previously by others by addition of a small molecule inhibitor of GSK3β, which was shown to increase NK cell maturation and cytotoxic activity (Cichocki et al, 2017; reference 5). The aim of the present manuscript was to study whether GSK3β itself caused these effects or whether these were off-target effects of the small molecule inhibitor. Therefore, the authors studied the effects of GSK3β gene deletion, but unfortunately they used an NK cell culture model in which there was no increase of GSK3β. Therefore, it is not unexpected that GSK3β gene deletion had no effect on proliferation, cytokine production or cytotoxicity. 

Altogether, this has a very negative impact on the intrinsic value of this manuscript.

Author response: Our apologies for appearing to be evasive- our intent was to clarify the goal of our study. We clarify for the reviewer that we never stated that our aim was to study GSK3β expression in IL-15 expanded NK cells or AML NK cells. That work had previously been done by others and we only reference that work as contextual background and for comparison. Our aim was to add to the literature regarding the effect of GSK3β in NK cells using a more refined and targeted approach (CRISPR knockout) than GSK3β inhibitors, and to identify the genes and pathways directly regulated by GSK3β. We thought it most relevant to use the expansion approach that we developed and were familiar with, as it has been widely implemented in clinical trials. Our findings did not match those of previous reports on GSK3β in NK cells, so we posited an explanation for our findings that included possible differences in our approach compared to the approaches used in these other papers.

To better address the concerns of the reviewer, we performed additional experiments using IL15-expanded NK cells. Since IL15 is not a focus of this manuscript, these experiments and their results have not been added to the manuscript but are presented for the reviewers at the end of this response letter. Briefly, we found that IL15- and IL21-expanded NK cells expressed similar levels of GSK3β at the protein and mRNA levels (Response Figure 1). Thus, although we did not confirm previous reports of IL-15-dependent overexpression of GSK3β, we do confirm robust expression of GSK3β in expanded NK cells, regardless of which cytokine is used during expansion. This suggests that differences in cytokine are not the explanation.

In addition, we attempted to use our gene editing approach to see if deletion of the GSK3B gene would restore the function of NK cells from AML patients, but we were unsuccessful due to the limited numbers, poor growth, and poor editing efficiency of these cells.

We assume the other responses made as comments in the manuscript were acceptable, and have marked them as resolved.

Reviewer 2: 2nd round comments

The authors have addressed the critiques adequately.

Author response: Thank you for your review.

Reviewer 3: 1st round comments

The main objective of this paper is to decipher the role of GSK3B in NK cell functions in patients with acute myeloblastic leukemia. Indeed, GSK3B is elevated in NK cells from AML patients and drug inhibition of GSK3B improves their cytotoxic activity.

This work includes the development of a GSK3B KO with Cas9/RNP targetting exon5, valided at both RNA and protein level.

With an in-vitro model, the transcriptomic analysis shows a limited set of differencially expressed genes beetwen WT and GSK3B-KO, mainly related to NK cell effector function. Among 12 downregulated genes in GSK3B-KO, the authors identified 2 genes involved in mitochondrial function and suggested a role of GSK3B in metabolism. Unfortunately, they did not show any difference on NK cell phenotype, cytotoxicity or cytokine production, altering the global message of this paper.

They did not compared the transcriptomic profile of NK cells from healthy donnor vs AML patients, but only pointed the expression of the genes selected from their in-vitro experiments.

Lastly, the conclusions on the role of GSK3B in AML patients are overinterpreted.

This work is nicely conducted and includes many interesting experiments, but the main message is not well supported by the functional analyses.

Minor revision:  The order and legends of figures are sometimes approximative.  

Author response: Thank you for these overall review comments. As mentioned in our response to Reviewer 1, our objective was not to better understand NK cells in AML. Rather, we sought to better elucidate the role of GSK3B in NK cells based on the previous reports in which the impact of IL15 and AML on NK cell function were a focus, and GSK3B was identified as a mechanism.

We have not made any new conclusions about the role of GSK3B in NK cells of patients with AML, other than a potential link between the two studies implicating genes related to metabolism.

We have further softened “confirming clinical relevance” to “confirming this correlation in a clinical setting” (line 31) and softened “shows a possible role” to “suggests a possible role” (line 375).

To further address the concerns of the reviewers we have more clearly stated our main objectives, consistent with the data we present here and restricted to the model we are using (line 14).

Please find enclosed the paper with our points of discussion:

Line 70 - Already mentioned in the patient sample section

Author response: Thank you for pointing this out, as we had not made this section clear. The methods used by our co-authors for isolation of AML-NK cells and control NK cells were different from the methods we used for performing CRISPR editing. We have modified this section to be more clear.

Line 72 - This kit provides a purification of NK cells by negative selection, not with positive markers as evocated below (CD56pos). Did you check the purity of NK sorted cells by flow cytometry with CD3/CD56/CD16 profiles. Please provide the mean purity of sorted cells and their phenotype.

Author response: Thank you for your comment. We have corrected this section.

Line 170 - There is no Figure 1C. Figure 1B with genome browser picture is less informative than box-plot, western blot could be interesting in the main figure 1.

Author response: Thank you for your comment. We had previously moved 1C to the supplementary figure following the editor’s suggestion, and missed this in the editing. Per your recommendation we have added the WB figure back and corrected the figure labels.

Line 178 - There is no IL-15 in your culture model, that could partially explain the absence of cytotoxicity difference in your study.

Moreover, drug inhibition of GSK3B could have off target effect on others GSKs explaining the absence of effects on your KO-NK cells. Did you test the GSK3B inhibitor in your model to evaluate this hypothesis and try to reproduce the previous results.

Unfortunately, this observation deserves the global message of your paper.

Author response:  We thank you for this comment. Please see results of new experiments addressing this issue, below at the end of this reviewer response. In summary, we did not find a difference in GSK3B expression between FC15 and FC21-expanded cells. Moreover, the inhibitor had a similar effect on the CD57/NKG2A/NKG2C phenotype of FC15-expanded NK cells, but did not augment their cytotoxic capacity. In addition, we also point out that the only difference between the Stage 5 and Stage 6 subsets is CD57 expression, but that there was no association between CD57 and GSK3B expression in these subsets (Figure 5). This suggests that there are functional differences between cytokine and feeder cell expansion methods, and that there may be off-target effects of GSK3β inhibitors, but we do not feel that we have sufficient data to support discussing this in the paper.

Line 192 - Have you analyzed the phenotype of your expanded NK cells by flow. An illustrative case could be interesting in this figure.

Author response: Thank you for this comment. We have extensively analyzed and published the phenotype of the expanded cells elsewhere. To address this comment specifically regarding the CD57 maturation markers, we have added this data as a new supplementary figure #4. Noting that the only difference between the Stage 5 and Stage 6 subsets is CD57, we also added a sentence in the discussion noting the lack of association between these subsets and GSK3B.

Line 217 - Please be careful with the order of figures in the main text. Figure C not provided.

Author response: Thank you for your comment. We added back Figure 1C and carefully checked and fixed the order of our figures.

Furthermore, the read drop accross exon 6 is directly related with the experimental deletion of GSK3B gene but have no direct link with the beggining of your sentense.

Author response: Thank you for your comment. We changed the wording of this section.

Line 219 - GSEA analysis can be used with both upregulated and dowregulated gene list, with a ranked score, Why dis you focus on the 55 overexpressed genes? The underexpressed genes can be related to the same enriched pathway and could improve your results.

Author response: We thank you for catching this error. We used all genes with a corrected p value of ≤ 0.05, of which there were 55 genes in total. 12 were upregulated, and 43 were downregulated. We have corrected this in the text.

Line 229 - These gens are not highlighted neither in the volcano plot, neither in the heatmap. You stressed on MT-ND1 or NT-MD2 in the figure). Please explain the link?

Author response: Thank you for your comment. For Fig 3A, the software picked several genes to highlight with name labels that have high DEG scores. For Fig 3C, all of the downregulated genes, including MT.ND1, 2, 3, 4, and 5, were included in the original heatmap. We inadvertently excluded 10 of the upregulated genes. Figure 3C has been revised to include all 55 DEGs.

Line 231 - please statuate on a positive or negative role on NK cell function ("regulate" an be confusing).

Author response: Thank you for your comment. We did not make this change, as the sentence refers broadly to 3 distinct areas of cellular function, for which individual genes may be upregulated or downregulated.

Line 236 - Please correct the figure legends: boxplot (D) heatmap (C). The legend

Author response: Thank you for pointing it out. We fixed this error.

Line 263 - Please explain the stage 5 and 6 selection in the section methods. They are related to CD56dim/CD16hi/CD57+/- subsets, are they FASC sorted or did you performed single-cell analysis?

Author response: We thank you for your comment- we appreciate that this was not explained thoroughly. We added explanatory text in the Methods and Materials, and referred the reader to the original paper for more detail.

Line 273 - You performed RNAseq on NK cells from healthy donnor and AML. You only stressed on the DEG obtained in the experimental culture model with GSK3B KO. Have you performed unsupervised analyses to compare NK cells from healthy donnor vs LAM? Can you provide this results perhaps in supplemental data and compare them to the expected results of your model.

Author response: Thank you for your comments. We did not perform clustering between GSK3B-KO NK cells and AML-NK. As mentioned above and in the manuscript, these experiments were performed at different centers using very different methods and on different NK cell subpopulations. Instead, we evaluated the same genes discovered from the in vitro KO set for validation/confirmation in primary human NK cells, using the control groups in each cohort as the comparator.

In order to properly address points raised by reviewer 1 and 3, we did some experiments listed here.

Question 1: Does expansion of NK cells with IL15-expressing feeder cells result in increased levels of GSK3b?

Purified NK cells were stimulated weekly for two weeks with irradiated Clone 4 feeder cells (K562 + 4-1BBL + membrane-bound IL15, referred to hereafter as FC15) in AIM-V expansion medium supplemented with ICSR and 50 IU of human recombinant IL-2 (rIL-2) to generate FC15 WT NK cells. We acknowledge that this model still different to the one presented by Cichocki et all – which utilized soluble IL15 to expand NK cells.

We also tested IL21-expanded WT, and IL21-expanded GSK3B-KO NK cells that we had generated for the manuscript from a different NK cell donor.

GSK3b expression was determined at the protein level by western blot, using the same method described in the manuscript. FC21-WT NK cells express high levels of GSK3b (first lane), which was undetectable in FC21-GSK3bKO NK cells (middle lane). FC15-WT NK cells (right lane) showed modestly higher levels of GSK3b, as quantified band densitometry, about ~60% higher when compared to FC21 GSK3bWT.

We had previously generated pairs of FC15- and FC21-expanded NK cells from 7 different donors, and performed RNAseq. We queried that data and show that there was no difference in GSK3B expression at the mRNA level between the two expansion approaches.

Round 2

Reviewer 1 Report

My first major comment was: “One of the aims of this study was to study whether the increased GSK3β expression in IL-15-expanded NK cells and in AML NK cells is the cause of their decreased maturation and cytotoxic antitumor activity, as previously indicated by the effects of a small molecule inhibitor of GSK3β. It is therefore unfortunate that a culture model was used here that did not show an increase in GSK3β expression in NK cells.”

The answer of the authors is: “We thank you for your comment. In the line 22-24, we clarified that FC-21 expanded NK cells were used as a model to study the biology of GSK due their stable expression of GSK after expansion. Using this model we could genetically engineer NK cells and we didn’t see any increase in the GSK3B level. Using models with elevated GSK3b such as IL15 expanded NK cells with would not be optimal to study the biology of these cells and it has also been studied by Miller lab at Cichocki et al, 2017.”

This is an evasive and irrelevant response. As indicated by the authors themselves in this manuscript, one of the aims was to study whether the increased GSK3β expression in IL-15-expanded NK cells and in NK cells from AML patients is causing their decreased maturation and cytotoxic antitumor activity. Evidence for this was provided previously by others by addition of a small molecule inhibitor of GSK3β, which was shown to increase NK cell maturation and cytotoxic activity (Cichocki et al, 2017; reference 5). The aim of the present manuscript was to study whether GSK3β itself caused these effects or whether these were off-target effects of the small molecule inhibitor. Therefore, the authors studied the effects of GSK3β gene deletion, but unfortunately they used an NK cell culture model in which there was no increase of GSK3β. Therefore, it is not unexpected that GSK3β gene deletion had no effect on proliferation, cytokine production or cytotoxicity.

Altogether, this has a very negative impact on the intrinsic value of this manuscript.

Author Response

Author response: Our apologies for appearing to be evasive- our intent was to clarify the goal of our study. We clarify for the reviewer that we never stated that our aim was to study GSK3β expression in IL-15 expanded NK cells or AML NK cells. That work had previously been done by others and we only reference that work as contextual background and for comparison. Our aim was to add to the literature regarding the effect of GSK3β in NK cells using a more refined and targeted approach (CRISPR knockout) than GSK3β inhibitors, and to identify the genes and pathways directly regulated by GSK3β. We thought it most relevant to use the expansion approach that we developed and were familiar with, as it has been widely implemented in clinical trials. Our findings did not match those of previous reports on GSK3β in NK cells, so we posited an explanation for our findings that included possible differences in our approach compared to the approaches used in these other papers.

To better address the concerns of the reviewer, we performed additional experiments using IL15-expanded NK cells. Since IL15 is not a focus of this manuscript, these experiments and their results have not been added to the manuscript but are presented for the reviewers at the end of this response letter. Briefly, we found that IL15- and IL21-expanded NK cells expressed similar levels of GSK3β at the protein and mRNA levels (Response Figure 1). Thus, although we did not confirm previous reports of IL-15-dependent overexpression of GSK3β, we do confirm robust expression of GSK3β in expanded NK cells, regardless of which cytokine is used during expansion. This suggests that differences in cytokine are not the explanation.

In addition, we attempted to use our gene editing approach to see if deletion of the GSK3B gene would restore the function of NK cells from AML patients, but we were unsuccessful due to the limited numbers, poor growth, and poor editing efficiency of these cells.

We assume the other responses made as comments in the manuscript were acceptable, and have marked them as resolved.

Reviewer 2 Report

The authors have addressed the critiques adequately.

Author Response

Thank you for your review.
